# Adherence to Mediterranean diet and female urinary incontinence: Evidence from the NHANES database

Shiwang Xie[1], Zuyi Li[2], Qinyuan Yao[3], Yupei Zhang[4]*, Yuan Ou[5]*

1 Luyuan Community Health Service Center Department, School of Medicine, Shenzhen & Longgang District People's Hospital of Shenzhen, The Chinese University of Hong Kong, Shenzhen, Guangdong Province, China, 2 Postpartum Healthcare Department, Shenzhen Longgang District Maternity & Child Healthcare Hospital, Shenzhen, Guangdong Province, China, 3 Community Health Service Center Department, The Second Affiliated Hospital, School of Medicine, Shenzhen & Longgang District People's Hospital of Shenzhen, The Chinese University of Hong Kong, Shenzhen, Guangdong Province, China, 4 Gynecology of Integrated Traditional Chinese and Western Medicine Department, The Second Affiliated Hospital, School of Medicine, Shenzhen & Longgang District People's Hospital of Shenzhen, The Chinese University of Hong Kong, Shenzhen, Guangdong Province, China, 5 Gynecology Department, Guangzhou Women and Children's Medical Center Liuzhou Hospital, Liuzhou, Guangxi Zhuang Autonomous Region, China

* yuanou78@outlook.com (YO); yuanou78@outlook.com (YZ)

**Data Availability Statement:** All data generated or analyzed during this study are available from the NHANES, https://www.cdc.gov/nchs/nhanes/about_nhanes.htm.

## Abstract

### Background

Urinary incontinence (UI) is a common condition in female. Oxidative stress and inflammation levels play important roles in UI progression. Mediterranean diet (MD) as a healthy anti-inflammatory dietary pattern has been reported to be associated with several inflammatory diseases. This study aimed to assess the association between the adherence to Mediterranean diet (aMED) and female UI.

### Methods

Data of study women aged ≥18 years old and diagnosed as stress UI and urgency UI were extracted from the National Health and Nutrition Examination Survey (NHANES) 2005–2018. Dietary intake information was obtained by 24-h dietary recall interview. Covariates included sociodemographic information, physical examination, and history of diseases and medication were extracted from the database. The weighted univariable and multivariate logistic regression models were used to assess the association between aMED and different types of UI, with odds ratios (ORs) and 95% confidence intervals (CIs). Subgroup analysis were further evaluated this association based on different age, body mass index (BMI), neutrophil to lymphocyte ratio (NLR), depression and smoking.

### Results

Totally, 13,291 women were included, of whom 5,921 (44.55%) had stress UI, 4276 (32.17%) had urgency UI and 2570 (19.34%) had mixed UI. After adjusted all covariates, high aMED score was associated with the lower odds of urgency (OR = 0.86, 95%CI: 0.75–

**Funding:** The author(s) received no specific funding for this work.

**Competing interests:** The authors have declared that no competing interests exist.

0.98) and mixed UI (OR = 0.84, 95%CI: 0.70–0.99), especially in female, aged 45–60 years old, NLR $\geq$1.68 and had smoking history. No relationship was found between the aMED and stress UI ($P$ >0.05).

## Conclusion

Greater aMED was connected with the low odds of urgency UI and mixed UI among female. Adherence to an anti-inflammatory diet in daily life are a promising intervention to be further explored in female UI.

## Introduction

Urinary incontinence (UI) is a common condition that can be divided into stress UI, urgency UI and mixed UI according to clinical manifestations, affecting approximately 25%-45% female worldwide [1, 2]. UI diminishes the quality of life for female and has been associated with an estimated $20 billion in annual direct health-care costs in the United States [3, 4]. Before breakthroughs are made in prevention or treatment, UI will pose an increasing challenge to the health care system around the world [5].

Pathophysiologic mechanisms of UI generally have been theorized to be related to inflammation level and oxidative stress [6, 7]. A review of Wu et al. [8] reported that overactive bladder (OAB) associated with inflammation can result from an imbalance between the production of pro-oxidants and their elimination through protective mechanisms of antioxidant-induced oxidative stress. Previous studies have suggested that dietary nutrients can promote oxidative stress and subsequently contribute to inflammation via cell signaling pathways [9–11]. Limited studies have found that higher intakes of energy and saturated fatty acids may be risk factors for UI in female, while certain antioxidant nutrients may be associated with a lower risk of UI in female [12, 13].

Mediterranean diet (MD), a classical anti-inflammatory dietary pattern which benefits on multiple aspects of human health, refers to a food profile characterized by high consumption of vegetables, fruits, legumes, nuts, moderately high consumption of fish, low consumption of dairy and meat products [14]. Among the components of MD, especially omega-3s, antioxidants and dietary fiber may help suppress inflammation in the body through different mechanisms [15–17]. Bozkurt et al. [18] reported that Med was related to the OAB and should be recommended in the first-line treatment of patients with OAB symptoms. Less is known, however, the association between adherence to Mediterranean diet (aMED) and female UI.

Herein, we explored the association between aMED and female UI, using the data from the National Health and Nutrition Examination Survey (NAHNES). This study aims to lay a theoretical foundation for the prevention and treatment of female UI from the perspective of improving diet habits.

## Methods

### Study design and participants

Data of seven consecutive survey cycles in this cross-sectional study were extracted from the NHANES 2005–2018. NHANES is a multipurpose survey conducted by the National Centers for Health Statistics (NCHS), a part of the Centers for Disease Control and Prevention (CDC) and assess the health and nutritional status of adults and children across the United States

[19]. Ethics approval was obtained from the NCHS Ethics Review Board and written informed consent was obtained from participants aged 12 years and older. Parental consent was obtained for those younger than 18 years. According to the Ethics Review Board of Guangzhou Women and Children's Medical Center Liuzhou Hospital, cross-sectional studies have been exempted from the ethical review.

The inclusion criteria were: (1) female aged ≥18 years; (2) female with the diagnosis of stress UI and urgency UI; (3) female with the complete information on dietary intake. The exclusion criteria: (1) pregnant women; (2) female with the history of bladder cancer, cervical cancer, uterine cancer and brain cancer; (3) missing number of vaginal deliveries.

## UI assessment

UI was assessed by four questions which each question had a binary response option (yes or no) in the kidney condition questionnaire [9, 10]. If participants answered yes to the question, "During the past 12 months, have you leaked or lost control of even a small amount of urine with activity like coughing, lifting or exercise?", she was defined as stress UI. If participants answered yes to the question, "During the past 12 months, have you leaked or lost control of even small amount of urine with an urge of pressure to urinate and you could not get to the toilet fast enough?", she was defined as urgency UI. Mixed UI was defined as having a positive response to both stress and urgency UI [20].

## aMED score assessment

Dietary nutrients data of all participants were obtained by 24-h dietary recall interview. The 24-h dietary recall interviews were conducted by face-to-face communication at the Mobile Examination Center (MEC). All participants were asked to recall the types and amount of food and drink consumed in the 24 h prior to the interview, while the use of dietary supplements were also recorded. We assessed our participants' adherence to the Mediterranean diet using the aMED score. The aMED score (total score = 18) are derived by an assigned value of "0", "1", or "2" across nine food categories (vegetables, legumes, fruits, nuts, whole grains, red and processed meats, fish, alcohol and olive oil), with higher scores indicating better adherence to MD pattern [21].

## Potential covariates

The potential covariates included age (years), race (Non-Hispanic White/Non- Hispanic Black/others), education level (less than high school/high school or equivalent/more than high school), marital status (married/never married/others), physical activity, poverty-to-income ratio (PIR) and body mass index (BMI). PIR was categorized as <1.3, 1.3–3.5, >3.5 and unknown [22]. Smoking status was assessed by the question 'Have you smoked at least 100 cigarettes in your entire life?' (yes/no) [23]. Drinking status (yes/no) [24] and marital status (married/never married/others) [25] were collected through personal interviews.

The medical history data adopted in this study were determined on the basis of the medical condition questionnaire (MSQ). Hypertension was defined as systolic blood pressure (SBP) ≥140 mmHg, diastolic blood pressure (DBP) ≥90 mmHg, self-reported high blood pressure or taking blood pressure medication [26]. Diabetes was defined as hemoglobin A1C (HbA1c) ≥6.5%, fasting glucose ≥126 mg/d L, 2 h oral glucose tolerance test (OGTT) blood glucose ≥200 mmol/L, self-reported diabetes history, or taking insulin or hypoglycemic agent [27]. Status of menopause was measured according to the following question 'Had regular periods in past 12 months?' (yes/no). If the participant answered the question negatively, then she was considered as menopause [28]. Number of vaginal deliveries was categorized as 0, 1 and ≥2

according to the question of 'How many vaginal deliveries'. Hysterectomy was measured according to the question of 'Had a hysterectomy?'. Neutrophil to lymphocyte ratio (NLR) refers to the ratio of neutrophil to Lymphocyte. Total energy and caffeine intake were calculated as the sum of dietary and supplements. Depression was tested by the Patient Health Questionnaire (PHQ-9), a 9-item screening instrument that asked the frequency of depression symptoms in the past two weeks [29]. The total score of PHQ-9 was 0–27 points, of which 0–9 was no depression, 10–14 was moderate depression, 15–19 was moderately severe depression, and 20–27 was severe depression. In present study, participants who scored ≥10 was considered as depression. Oral sex hormones use was assessed by the question '" Ever taken birth control pills" and "Ever use female hormones" [30].

## Statistical analysis

Continuous data were expressed as mean and standard error (S.E.), and the weighted t-test was used for comparison between groups. Categorical data were described by the number of cases and percentage [n (%)], and the chi-square test was used for comparison between groups. Missing data imputation was conducted using multivariate imputation by chained equations (MICE). Sensitivity analyses were performed to compare whether the results were different before and after imputation (S1 Table). The weighted univariate and multivariable logistic regression models were used to assess the associations between aMED and UI among female, with odds ratios (ORs) and 95% confidence intervals (CIs). Model I adjusted for age and race. Model II adjusted age, race, marital status, smoking, PIR, drinking, physical activity, number of vaginal deliveries, status of menopause, hysterectomy, diabetes, hypertension, depression, BMI, total energy, caffeine intake, moisture, NLR and oral sex hormones. Subgroup analysis were conducted to further assess this association based on different age, BMI, NLR, depression and smoking. Two-sided $P <0.05$ was considered statistically significant.

## Results

### Characteristics of study female

The flow chart of population screening was shown in Fig 1. Totally, 17,504 females were screened. Among them, 666 females missing the aMED assessment information, 617 pregnant females, 379 females with bladder cancer, cervical cancer, uterine cancer or brain cancer history, and 2,551 females missing vaginal deliveries numbers were excluded. Finally, 13,291 eligible females were included, with the mean age of 51.59 (0.23) years and the mean BMI was 29.48 (0.12) kg/m$^2$. Of whom, 5,921 (44.55%) had stress UI, 4276 (32.17%) had urgency UI and 2570 (19.34%) had mixed UI. Characteristics of included participants were shown in Table 1. Differences were found in age, race, the level of education, PIR, BMI, NLR and physical activity, marital, smoking and menopause status, number of vaginal, the history of hysterectomy, diabetes, hypertension and depression between two groups (all $P <0.05$).

### Association between aMED and female UI

Table 2 shows the association between aMED and different types of UI. After adjustment for age, race, education level, marital status, smoking, PIR, physical activity, vaginal deliveries numbers, status of menopause, hysterectomy, diabetes, hypertension, depression, BMI and NLR in model II, we observed that high aMED score was associated with the lower odds of urgency UI (OR = 0.86, 95%CI: 0.75–0.98) and mixed UI (OR = 0.84, 95%CI: 0.70–0.99). After adjustment for all above variables and drinking, total energy, caffeine intake, moisture and

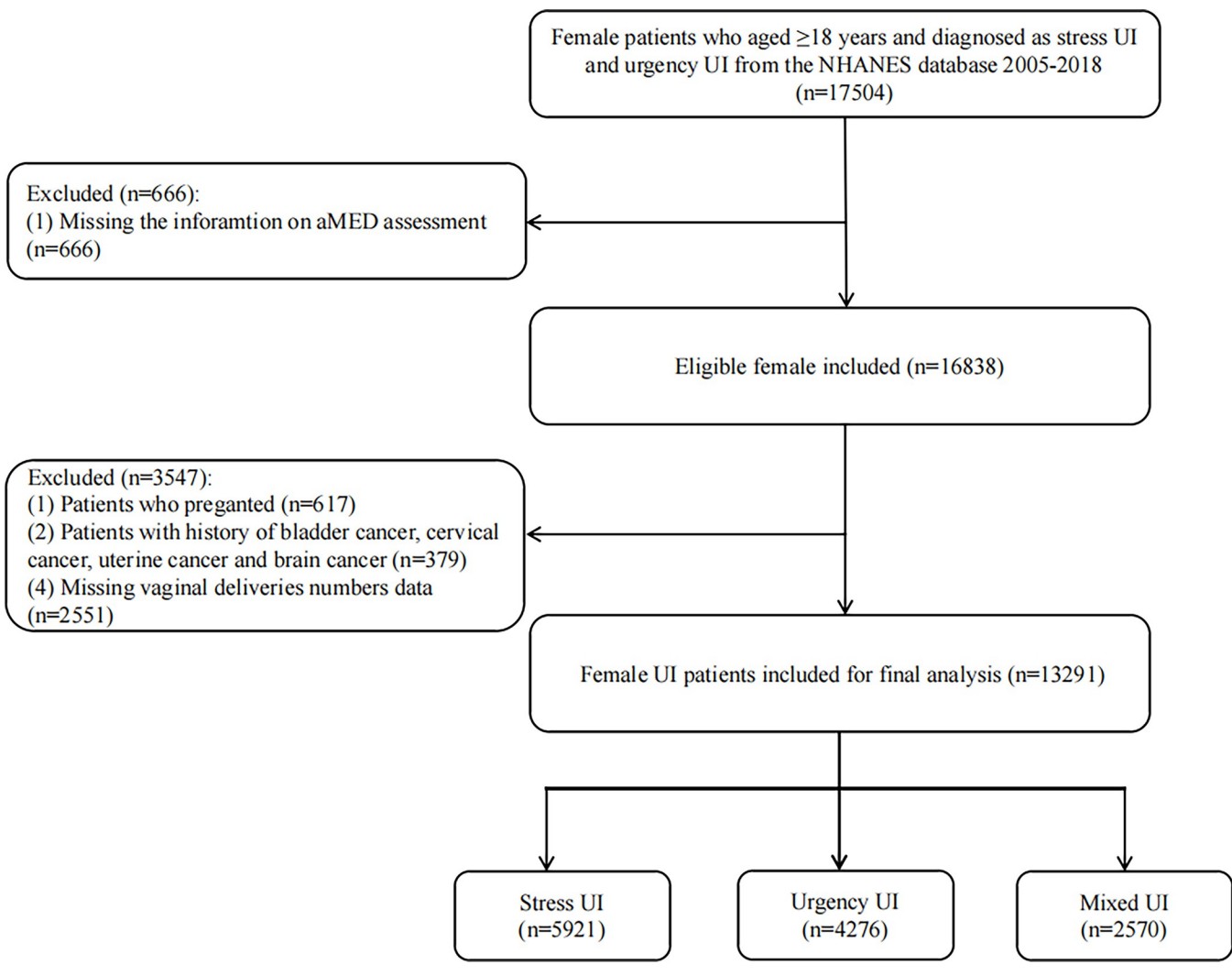

**Fig 1. The flow chart of population screening.**

oral sex hormones, no relationship were found between aMED score and stress UI in fully adjusted model ($P >0.05$).

## Association between aMED and female UI stratified by age, NLR, BMI, depression and smoking

The association between aMED and female UI stratified by age, NLR, BMI, depression and smoking were shown in Table 3. When stratified by age, high aMED score was associated with the lower odds of mixed UI (OR = 0.71, 95%CI: 0.51–0.97) among female (<45 years). In middle age female (45≤ age <60), high aMED score was associated with the lower odds of urgency UI (OR = 0.78, 95%CI: 0.62–0.98) and mixed UI (OR = 0.72, 95%CI: 0.54–0.98). No relationship was found between aMED score and UI among elderly female (≥ 60 years old). When stratified by NLR, high aMED score was associated with the low odds of urgency UI (OR = 0.74, 95%CI: 0.62–0.89) and mixed UI (OR = 0.79, 95%CI: 0.64–0.97) in NLR >1.68 group. When stratified by BMI, high aMED score was associated with the lower odds of mixed

**Table 1. Characteristics of study female.**

| Variates | Total (n = 13291) | Non-stress UI (n = 7370) | Stress UI (n = 5921) | P | Non-urgency UI (n = 9015) | Urgency UI (n = 4276) | P | No-mixed UI (n = 10721) | Mixed UI (n = 2570) | P |
|---|---|---|---|---|---|---|---|---|---|---|
| Med scores, n (%) | | | | 0.359 | | | 0.121 | | | 0.068 |
| ≤4 | 3538 (28.62) | 1936 (27.93) | 1602 (29.40) | | 2429 (28.18) | 1109 (29.62) | | 2849 (28.03) | 689 (31.19) | |
| 5–6 | 5546 (40.97) | 3105 (41.13) | 2441 (40.78) | | 3726 (40.63) | 1820 (41.73) | | 4473 (41.06) | 1073 (40.57) | |
| ≥7 | 4207 (30.41) | 2329 (30.94) | 1878 (29.81) | | 2860 (31.18) | 1347 (28.65) | | 3399 (30.91) | 808 (28.24) | |
| Age, years, Mean (S.E) | 51.59 (0.23) | 49.75 (0.29) | 53.69 (0.28) | <0.001 | 48.70 (0.24) | 58.23 (0.39) | <0.001 | 50.18 (0.25) | 57.74 (0.40) | <0.001 |
| Age, years, n (%) | | | | <0.001 | | | <0.001 | | | <0.001 |
| <45 | 4619 (35.15) | 2940 (41.41) | 1679 (27.99) | | 3790 (42.14) | 829 (19.06) | | 4124 (38.92) | 495 (18.61) | |
| 45–60 | 3617 (32.49) | 1724 (28.46) | 1893 (37.11) | | 2451 (32.31) | 1166 (32.91) | | 2827 (31.71) | 790 (35.91) | |
| ≥60 | 5055 (32.36) | 2706 (30.13) | 2349 (34.90) | | 2774 (25.55) | 2281 (48.02) | | 3770 (29.36) | 1285 (45.48) | |
| Race, n (%) | | | | <0.001 | | | <0.001 | | | 0.024 |
| Non-Hispanic White | 5550 (67.80) | 2767 (63.98) | 2783 (72.16) | | 3672 (67.21) | 1878 (69.16) | | 4363 (67.23) | 1187 (70.30) | |
| Non-Hispanic Black | 3025 (12.25) | 2043 (15.38) | 982 (8.67) | | 1952 (11.59) | 1073 (13.76) | | 2541 (12.62) | 484 (10.63) | |
| Others | 4716 (19.95) | 2560 (20.64) | 2156 (19.17) | | 3391 (21.20) | 1325 (17.09) | | 3817 (20.16) | 899 (19.07) | |
| Education level, n (%) | | | | 0.999 | | | <0.001 | | | <0.001 |
| Less than high school | 3392 (16.66) | 1848 (16.68) | 1544 (16.65) | | 2179 (15.71) | 1213 (18.85) | | 2607 (15.89) | 785 (20.04) | |
| High school graduation | 3099 (24.62) | 1740 (24.62) | 1359 (24.63) | | 2067 (23.63) | 1032 (26.90) | | 2479 (23.96) | 620 (27.52) | |
| More than high school | 6800 (58.71) | 3782 (58.71) | 3018 (58.72) | | 4769 (60.65) | 2031 (54.24) | | 5635 (60.14) | 1165 (52.44) | |
| Marital status, n (%) | | | | <0.001 | | | <0.001 | | | <0.001 |
| Married | 6708 (57.65) | 3558 (55.75) | 3150 (59.82) | | 4802 (60.14) | 1906 (51.91) | | 5528 (58.89) | 1180 (52.21) | |
| Never married | 1373 (8.17) | 922 (9.94) | 451 (6.14) | | 975 (8.41) | 398 (7.59) | | 1168 (8.47) | 205 (6.82) | |
| Others | 5210 (34.18) | 2890 (34.31) | 2320 (34.04) | | 3238 (31.44) | 1972 (40.50) | | 4025 (32.63) | 1185 (40.97) | |
| PIR, n (%) | | | | 0.050 | | | <0.001 | | | <0.001 |
| ≤1.3 | 4091 (21.95) | 2305 (22.21) | 1786 (21.66) | | 2708 (21.40) | 1383 (23.22) | | 3204 (21.21) | 887 (25.20) | |
| 1.3–3.5 | 4651 (33.65) | 2562 (33.46) | 2089 (33.87) | | 3110 (32.59) | 1541 (36.09) | | 3742 (32.99) | 909 (36.54) | |
| >3.5 | 3417 (37.65) | 1838 (36.88) | 1579 (38.54) | | 2447 (39.38) | 970 (33.69) | | 2863 (39.05) | 554 (31.53) | |
| Unknown | 1132 (6.74) | 665 (7.45) | 467 (5.93) | | 750 (6.63) | 382 (7.00) | | 912 (6.74) | 220 (6.72) | |
| Smoking, n (%) | | | | 0.002 | | | 0.017 | | | 0.003 |
| No | 8264 (59.08) | 4737 (60.99) | 3527 (56.90) | | 5728 (60.09) | 2536 (56.77) | | 6769 (60.04) | 1495 (54.90) | |
| Yes | 5027 (40.92) | 2633 (39.01) | 2394 (43.10) | | 3287 (39.91) | 1740 (43.23) | | 3952 (39.96) | 1075 (45.10) | |
| Drinking, n (%) | | | | 0.003 | | | 0.083 | | | 0.116 |
| No | 5212 (30.97) | 3003 (32.41) | 2209 (29.33) | | 3483 (30.40) | 1729 (32.31) | | 4160 (30.61) | 1052 (32.59) | |
| Yes | 8079 (69.03) | 4367 (67.59) | 3712 (70.67) | | 5532 (69.60) | 2547 (67.69) | | 6561 (69.39) | 1518 (67.41) | |
| PA, met·min/week, n (%) | | | | 0.080 | | | <0.001 | | | <0.001 |
| <450 | 1964 (15.40) | 1074 (14.79) | 890 (16.10) | | 1317 (15.10) | 647 (16.10) | | 1573 (15.01) | 391 (17.14) | |
| ≥450 | 6800 (55.48) | 3825 (56.76) | 2975 (54.01) | | 4893 (58.69) | 1907 (48.08) | | 5643 (57.25) | 1157 (47.69) | |
| Unknown | 4527 (29.12) | 2471 (28.45) | 2056 (29.89) | | 2805 (26.21) | 1722 (35.82) | | 3505 (27.74) | 1022 (35.17) | |
| Vaginal deliveries numbers, n (%) | | | | <0.001 | | | <0.001 | | | <0.001 |
| 0 | 2470 (19.59) | 1617 (23.83) | 853 (14.75) | | 1874 (21.54) | 596 (15.11) | | 2155 (21.10) | 315 (12.97) | |
| 1 | 2275 (18.73) | 1330 (19.72) | 945 (17.60) | | 1648 (19.98) | 627 (15.86) | | 1916 (19.51) | 359 (15.32) | |
| ≥2 | 8546 (61.68) | 4423 (56.45) | 4123 (67.65) | | 5493 (58.48) | 3053 (69.03) | | 6650 (59.39) | 1896 (71.70) | |

(*Continued*)

**Table 1.** (Continued)

| Variates | Total (n = 13291) | Non-stress UI (n = 7370) | Stress UI (n = 5921) | P | Non-urgency UI (n = 9015) | Urgency UI (n = 4276) | P | No-mixed UI (n = 10721) | Mixed UI (n = 2570) | P |
|---|---|---|---|---|---|---|---|---|---|---|
| Menopause status, n (%) | | | | <0.001 | | | <0.001 | | | <0.001 |
| No | 5448 (42.56) | 3242 (46.36) | 2206 (38.21) | | 4367 (49.64) | 1081 (26.26) | | 4778 (46.21) | 670 (26.57) | |
| Yes | 7843 (57.44) | 4128 (53.64) | 3715 (61.79) | | 4648 (50.36) | 3195 (73.74) | | 5943 (53.79) | 1900 (73.43) | |
| Hysterectomy, n (%) | | | | <0.001 | | | <0.001 | | | <0.001 |
| No | 9986 (75.71) | 5696 (78.20) | 4290 (72.85) | | 7184 (80.23) | 2802 (65.31) | | 8331 (78.47) | 1655 (63.58) | |
| Yes | 3305 (24.29) | 1674 (21.80) | 1631 (27.15) | | 1831 (19.77) | 1474 (34.69) | | 2390 (21.53) | 915 (36.42) | |
| Diabetes, n (%) | | | | <0.001 | | | <0.001 | | | <0.001 |
| No | 10608 (84.55) | 6071 (87.41) | 4537 (81.28) | | 7531 (87.51) | 3077 (77.74) | | 8800 (86.53) | 1808 (75.90) | |
| Yes | 2683 (15.45) | 1299 (12.59) | 1384 (18.72) | | 1484 (12.49) | 1199 (22.26) | | 1921 (13.47) | 762 (24.10) | |
| Hypertension, n (%) | | | | <0.001 | | | <0.001 | | | <0.001 |
| No | 6617 (54.63) | 3935 (59.44) | 2682 (49.13) | | 5082 (60.62) | 1535 (40.85) | | 5689 (57.94) | 928 (40.13) | |
| Yes | 6674 (45.37) | 3435 (40.56) | 3239 (50.87) | | 3933 (39.38) | 2741 (59.15) | | 5032 (42.06) | 1642 (59.87) | |
| Depression, n (%) | | | | <0.001 | | | <0.001 | | | <0.001 |
| No | 11771 (89.68) | 6733 (92.16) | 5038 (86.85) | | 8255 (92.04) | 3516 (84.27) | | 9755 (91.68) | 2016 (80.93) | |
| Yes | 1520 (10.32) | 637 (7.84) | 883 (13.15) | | 760 (7.96) | 760 (15.73) | | 966 (8.32) | 554 (19.07) | |
| BMI, kg/m$^2$, Mean (S.E) | 29.48 (0.12) | 28.60 (0.14) | 30.49 (0.17) | <0.001 | 28.82 (0.14) | 31.00 (0.18) | <0.001 | 29.04 (0.12) | 31.44 (0.22) | <0.001 |
| Obese, kg/m$^2$, n (%) | | | | <0.001 | | | <0.001 | | | <0.001 |
| <30 | 7474 (59.80) | 4447 (64.65) | 3027 (54.24) | | 5433 (63.72) | 2041 (50.77) | | 6312 (62.42) | 1162 (48.29) | |
| ≥30 | 5817 (40.20) | 2923 (35.35) | 2894 (45.76) | | 3582 (36.28) | 2235 (49.23) | | 4409 (37.58) | 1408 (51.71) | |
| Total energy, kcal, Mean (S.E) | 1794.34 (9.62) | 1760.22 (12.13) | 1833.37 (13.38) | <0.001 | 1795.84 (11.48) | 1790.90 (17.68) | 0.815 | 1786.31 (10.43) | 1829.57 (23.28) | 0.089 |
| Caffeine, mg, Mean (S.E) | 165.12 (3.07) | 153.99 (3.57) | 177.85 (4.59) | <0.001 | 164.11 (3.44) | 167.45 (4.71) | 0.516 | 163.26 (3.31) | 173.27 (6.25) | 0.137 |
| Moisture, gm, Mean (S.E) | 2754.23 (20.00) | 2683.63 (23.65) | 2834.99 (28.17) | <0.001 | 2762.14 (24.10) | 2736.04 (28.70) | 0.453 | 2742.85 (21.77) | 2804.12 (37.99) | 0.137 |
| NLR, Mean (S.E) | 2.14 (0.02) | 2.11 (0.02) | 2.17 (0.02) | 0.007 | 2.12 (0.02) | 2.18 (0.03) | 0.041 | 2.13 (0.02) | 2.20 (0.03) | 0.014 |
| NLR, n (%) | | | | 0.009 | | | 0.529 | | | 0.978 |
| <1.68 | 5202 (35.06) | 3000 (36.34) | 2202 (33.59) | | 3543 (34.82) | 1659 (35.61) | | 4245 (35.07) | 957 (35.02) | |
| ≥1.68 | 8089 (64.94) | 4370 (63.66) | 3719 (66.41) | | 5472 (65.18) | 2617 (64.39) | | 6476 (64.93) | 1613 (64.98) | |
| Oral sex hormones, n (%) | | | | <0.001 | | | 0.188 | | | 0.123 |
| No | 3388 (19.06) | 2058 (21.17) | 1330 (16.65) | | 2353 (19.41) | 1035 (18.27) | | 2782 (19.35) | 606 (17.82) | |
| Yes | 9903 (80.94) | 5312 (78.83) | 4591 (83.35) | | 6662 (80.59) | 3241 (81.73) | | 7939 (80.65) | 1964 (82.18) | |

S.E: standard error; Med: Mediterranean; UI: urinary incontinence; PA: physical activity; PIR: poverty-to-income ratio; BMI: body mass index; met: metabolic equivalent task; NLR: neutrophil to lymphocyte ratio.

UI in BMI <30 kg/m$^2$ group (OR = 0.77, 95%CI: 0.60–0.99). No relationship was found between aMED score and female UI with BMI ≥30 kg/m$^2$. When stratified by depression, high aMED score was related to the lower odds of urgency UI (OR = 0.86, 95%CI: 0.75–0.99) among patients without depression. No relationship was found between aMED score and UI among female with depression. When stratified by smoking, high aMED score was related to the lower odds of urgency UI (OR = 0.76, 95%CI: 0.63–0.92) and mixed UI (OR = 0.74, 95% CI: 0.59–0.94) among smoked participants.

**Table 2. Association between aMED and differences types of UI.**

| Variables | Model I | | Model II | |
|---|---|---|---|---|
| | OR (95%CI) | *P* | OR (95%CI) | *P* |
| Stress UI | | | | |
| aMED | | | | |
| ≤4 | Ref | | Ref | |
| 5–6 | 0.94 (0.86–1.04) | 0.250 | 0.98 (0.88–1.08)* | 0.631 |
| ≥7 | 0.90 (0.79–1.03) | 0.119 | 0.96 (0.84–1.11) * | 0.587 |
| Urgency UI | | | | |
| aMED | | | | |
| ≤4 | Ref | | Ref | |
| 5–6 | 0.91 (0.79–1.05) | 0.179 | 0.91 (0.79–1.04)! | 0.167 |
| ≥7 | 0.77 (0.68–0.88) | <0.001 | 0.86 (0.75–0.98) ! | 0.020 |
| Mixed UI | | | | |
| aMED | | | | |
| ≤4 | Ref | | Ref | |
| 5–6 | 0.84 (0.71–1.00) | 0.052 | 0.84 (0.71–0.99)# | 0.047 |
| ≥7 | 0.75 (0.63–0.89) | 0.001 | 0.84 (0.70–0.99) # | 0.048 |

Ref: reference; OR: odds ratio; CI: confidence interval.

aMED: adherence to Mediterranean diet; UI: urinary incontinence.

Model I: crude model

In model II

*: adjusted for age, race, marital status, smoking, PIR, drinking, physical activity, number of vaginal deliveries, status of menopause, hysterectomy, diabetes, hypertension, depression, BMI, total energy, caffeine intake, moisture, NLR and oral sex hormones

!: adjusted for age, race, education level, marital status, smoking, PIR, physical activity, number of vaginal deliveries, status of menopause, hysterectomy, diabetes, hypertension, depression, BMI and NLR

#: adjusted for age, race, education level, marital status, smoking, PIR, physical activity, number of vaginal deliveries, status of menopause, hysterectomy, diabetes, hypertension, depression, BMI and NLR.

## Discussion

We investigated the association between aMED and stress UI, urgency UI and mixed UI among female in present study. Our results showed that high aMED was associated with the lower odds of urgency UI and mixed UI. No association was found between aMED and stress UI among generally female.

UI was divided into stress, urgency and mixed UI. Stress UI was the leakage of urine to increased intra-abdominal pressure such as exercise and cough, which is due to the poor functional urethra. Urgency UI refers to the reduction of anatomical support due to trauma, vaginal delivery, obesity and increased intra-abdominal pressure due to chronic constipation. If both urgency UI and stress UI are present together, it is called a mixed UI [31]. Modification of behavioral and lifestyle remain the preferred first line of treatment for most UI patients, however there have been few studies of dietary correlates of UI. However, there is increasing evidence that diet may play a significant role in the development of UI symptoms. A review conducted by Aragón et al. [32] suggested that some dietary factors, such as cranberry juice and fermented milk products, can reduce the risk of contracting urinary diseases by altering the properties of the urogenital bacterial floral. Another systemic review and meta-analysis summarized that vitamin D deficiency increases the risk of OAB and UI, while vitamin D supplementation reduces the risk of UI, which may due to the effect of vitamin D on muscle

**Table 3. Association between aMED and different types of UI stratified by age, NLR, BMI, depression and smoking.**

| Outcomes | Stress UI | | Urgency UI | | Mixed UI | |
|---|---|---|---|---|---|---|
| | OR (95%CI) | *P* | OR (95%CI) | *P* | OR (95%CI) | *P* |
| **Age<45 (n = 4619)** | | | | | | |
| n | 1679 (37.14%) | | 829 (16.43%) | | 495 (9.84%) | |
| aMED | | | | | | |
| ≤4 | Ref | | Ref | | Ref | |
| 5–6 | 0.94 (0.77–1.14)[a] | 0.503 | 0.81 (0.63–1.05)[f] | 0.109 | 0.74 (0.53–1.02) | 0.064 |
| ≥7 | 0.94 (0.78–1.13)[a] | 0.520 | 0.83 (0.65–1.07)[f] | 0.159 | 0.71 (0.51–0.97) | 0.033 |
| **45≤ Age <60 (n = 3617)** | | | | | | |
| n | 1893 (53.28%) | | 1166 (30.68%) | | 790 (20.53%) | |
| aMED | | | | | | |
| ≤4 | Ref | | Ref | | Ref | |
| 5–6 | 0.93 (0.72–1.20)[a] | 0.571 | 0.86 (0.67–1.12)[f] | 0.264 | 0.78 (0.57–1.05) | 0.101 |
| ≥7 | 0.85 (0.64–1.13)[a] | 0.262 | 0.78 (0.62–0.98)[f] | 0.036 | 0.72 (0.54–0.98) | 0.034 |
| **Age ≥ 60 (n = 5055)** | | | | | | |
| n | 2349 (50.32%) | | 2281 (44.95%) | | 1285 (26.12%) | |
| aMED | | | | | | |
| ≤4 | Ref | | Ref | | Ref | |
| 5–6 | 1.14 (0.93–1.39)[a] | 0.198 | 1.04 (0.83–1.30)[f] | 0.719 | 1.01 (0.77–1.33) | 0.934 |
| ≥7 | 1.14 (0.90–1.43)[a] | 0.270 | 0.97 (0.77–1.24)[f] | 0.819 | 1.07 (0.81–1.41) | 0.648 |
| **NLR<1.68 (n = 5202)** | | | | | | |
| n | 2202 (44.69%) | | 1659 (30.76%) | | 957 (18.56%) | |
| aMED | | | | | | |
| ≤4 | Ref | | Ref | | Ref | |
| 5–6 | 0.94 (0.77–1.14)[b] | 0.499 | 0.88 (0.68–1.13)[g] | 0.314 | 0.73 (0.54–0.98) | 0.038 |
| ≥7 | 0.95 (0.74–1.21)[b] | 0.655 | 1.09 (0.87–1.37)[g] | 0.452 | 0.93 (0.68–1.27) | 0.638 |
| **NLR≥1.68 (n = 8089)** | | | | | | |
| n | 3719 (47.71%) | | 2617 (30.03%) | | 1613 (18.59%) | |
| aMED | | | | | | |
| ≤4 | Ref | | Ref | | Ref | |
| 5–6 | 1.01 (0.89–1.14)[b] | 0.934 | 0.93 (0.79–1.08)[g] | 0.328 | 0.90 (0.75–1.09) | 0.272 |
| ≥7 | 0.98 (0.81–1.18)[b] | 0.816 | 0.74 (0.62–0.89)[g] | 0.002 | 0.79 (0.64–0.97) | 0.025 |
| **BMI<30kg/m² (n = 7474)** | | | | | | |
| n | 3027 (42.32%) | | 2041 (25.72%) | | 1162 (15.00%) | |
| aMED | | | | | | |
| ≤4 | Ref | | Ref | | Ref | |
| 5–6 | 0.97 (0.84–1.11)[c] | 0.634 | 0.97 (0.78–1.19)[h] | 0.740 | 0.86 (0.69–1.07) | 0.178 |
| ≥7 | 0.92 (0.76–1.12)[c] | 0.419 | 0.83 (0.69–1.01)[h] | 0.060 | 0.77 (0.60–0.99) | 0.047 |
| **BMI ≥30kg/m² (n = 5817)** | | | | | | |
| n | 2894 (53.09%) | | 2235 (37.09%) | | 1408 (23.90%) | |
| aMED | | | | | | |
| ≤4 | Ref | | Ref | | Ref | |
| 5–6 | 0.97 (0.82–1.15)[c] | 0.747 | 0.84 (0.68–1.04)[h] | 0.108 | 0.81 (0.64–1.01) | 0.062 |
| ≥7 | 1.00 (0.79–1.26)[c] | 0.999 | 0.88 (0.71–1.10)[h] | 0.261 | 0.90 (0.68–1.20) | 0.477 |
| **Non-Depression (n = 11771)** | | | | | | |
| n | 5038 (45.17%) | | 3516 (28.46%) | | 2016 (16.77%) | |
| aMED | | | | | | |
| ≤4 | Ref | | Ref | | Ref | |

*(Continued)*

**Table 3.** (Continued)

| Outcomes | Stress UI | | Urgency UI | | Mixed UI | |
|---|---|---|---|---|---|---|
| | OR (95%CI) | P | OR (95%CI) | P | OR (95%CI) | P |
| 5–6 | 0.97 (0.86–1.10)[d] | 0.653 | 0.92 (0.79–1.08)[i] | 0.310 | 0.86 (0.72–1.02) | 0.088 |
| ≥7 | 0.98 (0.85–1.14)[d] | 0.818 | 0.86 (0.75–0.99)[i] | 0.039 | 0.86 (0.71–1.05) | 0.133 |
| Depression (n = 1520) | | | | | | |
| n | 883 (59.47%) | | 760 (46.19%) | | 554 (34.35%) | |
| aMED | | | | | | |
| ≤4 | Ref | | Ref | | Ref | |
| 5–6 | 1.02 (0.72–1.44)[d] | 0.916 | 0.79 (0.55–1.15)[i] | 0.222 | 0.76 (0.50–1.15) | 0.189 |
| ≥7 | 0.81 (0.53–1.23)[d] | 0.322 | 0.83 (0.57–1.21)[i] | 0.337 | 0.72 (0.48–1.07) | 0.104 |
| Non-smoking (n = 8264) | | | | | | |
| n | 3527 (44.93%) | | 2536 (29.10%) | | 1495 (17.26%) | |
| aMED | | | | | | |
| ≤4 | Ref | | Ref | | Ref | |
| 5–6 | 1.04 (0.89–1.21)[e] | 0.622 | 1.02 (0.85–1.22)[j] | 0.815 | 0.96 (0.77–1.18) | 0.668 |
| ≥7 | 1.06 (0.90–1.24)[e] | 0.513 | 0.96 (0.80–1.15)[j] | 0.649 | 0.95 (0.75–1.20) | 0.643 |
| Smoking (n = 5027) | | | | | | |
| n | 2394 (49.14%) | | 1740 (32.00%) | | 1075 (20.48%) | |
| aMED | | | | | | |
| ≤4 | Ref | | Ref | | Ref | |
| 5–6 | 0.91 (0.76–1.08)[e] | 0.280 | 0.79 (0.65–0.95)[j] | 0.011 | 0.73 (0.56–0.94) | 0.016 |
| ≥7 | 0.86 (0.68–1.08)[e] | 0.196 | 0.76 (0.63–0.92)[j] | 0.005 | 0.74 (0.59–0.94) | 0.012 |

Ref: reference; OR: odds ratio; CI: confidence interval.

aMED: adherence to Mediterranean diet; UI: urinary incontinence.

[a]: adjusted for age, race, marital status, smoking, PIR, drinking, physical activity, number of vaginal deliveries, status of menopause, hysterectomy, diabetes, hypertension, depression, BMI, total energy, caffeine intake, moisture, NLR and oral sex hormones

[b]: adjusted for age, race, marital status, smoking, PIR, drinking, physical activity, number of vaginal deliveries, status of menopause, hysterectomy, diabetes, hypertension, depression, BMI, total energy, caffeine intake, moisture and oral sex hormones

[c]: adjusted for age, race, marital status, smoking, PIR, drinking, physical activity, number of vaginal deliveries, status of menopause, hysterectomy, diabetes, hypertension, depression, total energy, caffeine intake, moisture, NLR and oral sex hormones

[d]: adjusted for age, race, marital status, smoking, PIR, drinking, physical activity, number of vaginal deliveries, status of menopause, hysterectomy, diabetes, hypertension, BMI, total energy, caffeine intake, moisture, NLR and oral sex hormones

[e]: adjusted for age, race, marital status, PIR, drinking, physical activity, number of vaginal deliveries, status of menopause, hysterectomy, diabetes, hypertension, depression, BMI, total energy, caffeine intake, moisture, NLR and oral sex hormones

[f]: adjusted for age, race, education level, marital status, smoking, PIR, physical activity, number of vaginal deliveries, status of menopause, hysterectomy, diabetes, hypertension, depression, BMI and NLR

[g]: adjusted for age, race, education level, marital status, Smoke, PIR, physical activity, number of vaginal deliveries, status of menopause, hysterectomy, diabetes, hypertension, depression and BMI

[h]: adjusted for age, race, education level, marital status, smoking, PIR, physical activity, number of vaginal deliveries, status of menopause, hysterectomy, diabetes, hypertension, depression and NLR

[i]: adjusted for age, race, education level, marital status, smoking, PIR, physical activity, number of vaginal deliveries, status of menopause, hysterectomy, diabetes, hypertension, BMI and NLR

[j]: adjusted for age, race, education level, marital status, PIR, physical activity, number of vaginal deliveries, status of menopause, hysterectomy, diabetes, hypertension, depression, BMI and NLR.

growth and function, as well as decreasing the inflammatory response [33]. Dallosso et al. [34] collected data from 7046 women aged over 60 years to investigate the role of dietary and lifestyle in OAB and UI. The findings suggested that smoking, obesity and carbonated drinks

were confirmed to for bladder disorders associated with incontinence. The risk of OAB symptoms was reduced with higher consumption of vegetables, bread and chicken. In our study, significant relationships were observed between high MD score and the low odds of urgency UI and mixed UI among smoked participants, which were consistent with Dallosso's study. Another narrative review of John et al. also [35] reported that quitting smoking and health diet have resulted in a decreased incidence, prevalence and severity of UI. However, the association between smoking and UI remains weak so far. Some cross-sectional studies reported that tobacco could have a large impact on UI prevalence or play no role [36]. A comprehensive systemic review and meta-analysis concluded that smoking is the important factor influencing the incidence of UI in older women ($P<0.001$) [37]. This may be due to smoke result in bladder sphincter dysfunction, which leads to UI.

We also found high aMED score was associated with the low odds of UI among participants aged <60 years, while no relationship was found between aMED score and UI among participants aged ≥60 years. Indeed, a steady increase and severe UI has been reported throughout the adult lifespan, with a distinct peak around the menopausal age (around the fifth and the sixth decade) [38]. Estrogen can provide nutrients for the female urethral mucosa and increase the folds of the urethral mucosa, so the greater the pressure of the lumen, the less likely to leak urine. With the onset of perimenopause, the secretion of estrogen decreases significantly, the mucosal muscles of the urethra lose the nutrients of estrogen and atrophy, and the urethral mucosa decreases, thus increasing the chance of UI. A cross-sectional study of Rortveit et al. [39] reported that parity was an important risk factor for female UI in fertile and peri-and early postmenopausal ages. Only stress and mixed UI were related to parity and all effects of parity seem to disappear in older age. In addition, stress UI in fertile women was also related to a change in collagen metabolism resulting in an increasing concentration of collagen and large collagen fibrils.

We observed that greater aMED score was related to the low odds of stress and urgency UI among female participants who had high level of NLR. Previous studies have reported the association between inflammatory and OAB and urgency UI is a common symptom of OAB. A cross-sectional study conducted by Wei et al. [40] reported that systemic immunity-inflammation (SII) was associated with an 18% higher odds of OAB in the fully adjusted model. A cross-sectional survey of Zhang et al. [41] suggested that more pro-inflammatory diets, as presented by higher dietary inflammatory index scores (DII) are associated with an increased likelihood of UI in U.S. There were many possible explanations for the relationship between UI and DII score. Kobatake K et al. [42] indicated that inflammatory cytokines play a key role in the modulation of connexins expression and the pathogenesis of urinary bladder dysfunction.

Furthermore, several studies have focused the relationship between depression and dietary. A systematic review of Molendijk et al. [43] reported that high quality of diet was associated with lower odds for the onset of depressive symptoms. Another study also demonstrated that adopting an anti-inflammatory diet may be effective intervention or preventative means of reducing depression risk and symptoms [44]. However, in present study, no relationship was found between depression and aMED among UI female, this may due to the change in treatment and lifestyle of participants with depression and UI. Further long-term studies amongst different populations are required to explore the association between depression and aMED among UI participants.

Herein, the associations of aMED and different types of UI was explored based on the representative and high-quality NHANES database, and the regression models were adjusted considering the covariates influencing the UI. Moreover, we further explored these associations in several subgroup analyses to evaluate its robustness. In light of the above, our research further supports the importance of the MD as a potentially healthy eating pattern. MD is a dietary

style known for being healthy, simple, and nutritious. It is a dietary habit rather than a structured diet and emphasizes an adequate intake of fruits, vegetables, and whole grains and also contains moderate amounts of legumes, nuts, skim milk, olive oil and some fish, as well as small amounts of red meat, salt and carbohydrates [45]. Diet is actually a lifelong habit, therefore, greater attention should be given to the diet as a complex of bioactive compounds and nutrients, and their interaction effects. In this condition, aMED has emerged as the best behaviors to prevent several chronic diseases. For clinicians and policymakers, as well as UI patients, it was essential to be aware of the benefits of adherence to the MD for the health management of female UI. However, several limitations of this study can be identified. First, the information of dietary intake was obtained by 24-h dietary recall interview and the memory bias of participants may lead to a challenge to acquiring the accurate evaluation, especially for the elderly included in this study; moreover, the cross-sectional study design of this study could not establish a causal relationship between aMED and UI among female participants. Prospective large-scale studies are needed to further explore the association between Med score and UI among female participants.

## Conclusion

Greater aMED was associated with the low odds of urgency UI and mixed UI among female UI participants, especially in participants aged 45 to 60 years, NLR $\geq$1.68 and had the history of smoking. Future large-scale cohort studies are needed to explore the causal association between anti-inflammatory diet and female UI.

## Supporting information

**S1 Table. Sensitivity analyses for data before and after imputation.**
(DOCX)

## Author Contributions

**Conceptualization:** Shiwang Xie, Yuan Ou.

**Data curation:** Zuyi Li, Qinyuan Yao, Yupei Zhang.

**Formal analysis:** Zuyi Li, Qinyuan Yao, Yupei Zhang.

**Investigation:** Zuyi Li, Qinyuan Yao, Yupei Zhang.

**Methodology:** Zuyi Li, Qinyuan Yao, Yupei Zhang.

**Writing – original draft:** Shiwang Xie.

**Writing – review & editing:** Shiwang Xie, Yuan Ou.

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
