## [Decision Letter · Decision Letter 0]

25 Jul 2024

PONE-D-24-16803Adherence to Mediterranean diet and female urinary incontinence: evidence from the NHANES databasePLOS ONE

Dear Dr. Ou,

Thank you for submitting your manuscript to PLOS ONE. After careful consideration, we feel that it has merit but does not fully meet PLOS ONE’s publication criteria as it currently stands. Therefore, we invite you to submit a revised version of the manuscript that addresses the points raised during the review process.

We look forward to receiving your revised manuscript.

Kind regards,

Aleksandra Klisic

Academic Editor

PLOS ONE

Journal Requirements:

Reviewers' comments:

Reviewer's Responses to Questions

**Comments to the Author**

1. Is the manuscript technically sound, and do the data support the conclusions?

Reviewer #1: Yes

Reviewer #2: Yes

2. Has the statistical analysis been performed appropriately and rigorously? 

Reviewer #1: Yes

Reviewer #2: Yes

3. Have the authors made all data underlying the findings in their manuscript fully available?

Reviewer #1: Yes

Reviewer #2: No

4. Is the manuscript presented in an intelligible fashion and written in standard English?

Reviewer #1: Yes

Reviewer #2: Yes

5. Review Comments to the Author

Reviewer #1: The presented knowledge is not novel. However the conducted study demonstrates high scientific quality and is both necessary and relevant to the advancement of nutritional science.

1. The introduction is correctly presented, with the Authors fully justifying the need for such an analysis using the numerous studies available on the subject.

2. The study was conducted without methodological errors. The methods are described in detail and it is possible to replicate the study.

3. The description of the results is correct and in line with the usual description of manuscripts. The tables and figures presented are done correctly.

4. The Discussion section contained an accurate reference of the results obtained to the studies of other authors. However, I believe that other physiological factors that may significantly influence the condition under study have been discussed too generally. A broader discussion of the topic would definitely increase the scientific value of this paper.

5. The conclusions are well formulated and fully relate to the results obtained.

6. The formatting of the text and the layout of the manuscript are correct, according to the requirements of the journal.

Reviewer #2: Xie et al. have performed a study on the adherence to Mediterranean diet and female urinary incontinence using the NHANES data from 2005 to 2018. The study findings are interesting and the manuscript is well-written. These are my comments:

- Authors should add strengths to their manuscript before the limitation.

- A paragraph summarizing the clinical take-home message of this manuscript should be added to the discussion.

- The references prior to 2010 could be updated with those after 2010 since they provide more up-to-date findings.

6. PLOS authors have the option to publish the peer review history of their article (what does this mean?). If published, this will include your full peer review and any attached files.

Reviewer #1: No

Reviewer #2: No

---

## [Author Response · Author response to Decision Letter 0]

4 Aug 2024

Review Comments to the Author

Reviewer #1: The presented knowledge is not novel. However the conducted study demonstrates high scientific quality and is both necessary and relevant to the advancement of nutritional science.

1. The introduction is correctly presented, with the Authors fully justifying the need for such an analysis using the numerous studies available on the subject.

Reply: Thanks for your recognition for our study. 

2. The study was conducted without methodological errors. The methods are described in detail and it is possible to replicate the study.

Reply: Thanks for your recognition for our study.

3. The description of the results is correct and in line with the usual description of manuscripts. The tables and figures presented are done correctly.

Reply: Thanks for your recognition for our study.

4. The Discussion section contained an accurate reference of the results obtained to the studies of other authors. However, I believe that other physiological factors that may significantly influence the condition under study have been discussed too generally. A broader discussion of the topic would definitely increase the scientific value of this paper.

Reply: Thanks for your comment. According to your opinions, we checked our manuscript and the relevant physiological mechanisms were described in further detail. All changes were marked in red in the manuscript.

5. The conclusions are well formulated and fully relate to the results obtained.

Reply: Thanks for your recognition for our study.

6. The formatting of the text and the layout of the manuscript are correct, according to the requirements of the journal.

Reply: Thanks for your recognition for our study.

Reviewer #2: Xie et al. have performed a study on the adherence to Mediterranean diet and female urinary incontinence using the NHANES data from 2005 to 2018. The study findings are interesting and the manuscript is well-written. These are my comments:

- Authors should add strengths to their manuscript before the limitation.

Reply: Thanks for your comment. We have supplemented the strengths of our study in the Discussion section. 

- A paragraph summarizing the clinical take-home message of this manuscript should be added to the discussion.

Reply: Thanks for your comment. We have supplemented the clinical take-home information in the Discussion section. 

- The references prior to 2010 could be updated with those after 2010 since they provide more up-to-date findings.

Reply: Thanks for your comment. We have updated the references prior to 2010 in the manuscript. However, there were still several studies prior to 2010 that were very relevant to our study, so we did not replace these references.

---

## [Decision Letter · Decision Letter 1]

16 Sep 2024

Adherence to Mediterranean diet and female urinary incontinence: evidence from the NHANES database

PONE-D-24-16803R1

Dear Dr. Ou,

We’re pleased to inform you that your manuscript has been judged scientifically suitable for publication and will be formally accepted for publication once it meets all outstanding technical requirements.

Kind regards,

Aleksandra Klisic

Academic Editor

PLOS ONE

Additional Editor Comments (optional):

Reviewers' comments:

Reviewer's Responses to Questions

**Comments to the Author**

1. If the authors have adequately addressed your comments raised in a previous round of review and you feel that this manuscript is now acceptable for publication, you may indicate that here to bypass the “Comments to the Author” section, enter your conflict of interest statement in the “Confidential to Editor” section, and submit your "Accept" recommendation.

Reviewer #2: All comments have been addressed

Reviewer #3: (No Response)

2. Is the manuscript technically sound, and do the data support the conclusions?

Reviewer #2: (No Response)

Reviewer #3: (No Response)

3. Has the statistical analysis been performed appropriately and rigorously? 

Reviewer #2: (No Response)

Reviewer #3: (No Response)

4. Have the authors made all data underlying the findings in their manuscript fully available?

Reviewer #2: (No Response)

Reviewer #3: (No Response)

5. Is the manuscript presented in an intelligible fashion and written in standard English?

Reviewer #2: (No Response)

Reviewer #3: (No Response)

6. Review Comments to the Author

Reviewer #2: (No Response)

Reviewer #3: I comprehensively read the manuscript and evaluated the response from previous reviewers. The manuscript is acceptable.

7. PLOS authors have the option to publish the peer review history of their article (what does this mean?). If published, this will include your full peer review and any attached files.

Reviewer #2: No

Reviewer #3: No

---

## [Editor Report · Acceptance letter]

10 Oct 2024

PONE-D-24-16803R1 

PLOS ONE

Dear Dr. Ou, 

I'm pleased to inform you that your manuscript has been deemed suitable for publication in PLOS ONE. Congratulations! Your manuscript is now being handed over to our production team.

Kind regards, 

on behalf of

Dr. Aleksandra Klisic 

Academic Editor

PLOS ONE